# RNF149 Promotes HCC Progression through Its E3 Ubiquitin Ligase Activity

**DOI:** 10.3390/cancers15215203

**Published:** 2023-10-29

**Authors:** Zhaoyu Guo, Pei Jiang, Qian Dong, Yiming Zhang, Kaikun Xu, Yuanjun Zhai, Fuchu He, Chunyan Tian, Aihua Sun

**Affiliations:** 1State Key Laboratory of Proteomics, Beijing Proteome Research Center, National Center for Protein Sciences (Beijing), Beijing Institute of Lifeomics, Beijing 102206, China; guozhaoyu@ncpsb.org.cn (Z.G.); jiangpei@ncpsb.org.cn (P.J.); dongqian@ncpsb.org.cn (Q.D.); zhangyiming@ncpsb.org.cn (Y.Z.); xukaikun@ncpsb.org.cn (K.X.); zhaiyuanjun@ncpsb.org.cn (Y.Z.); hefc@bmi.ac.cn (F.H.); 2Research Unit of Proteomics Dirven Cancer Precision Medicine, Chinese Academy of Medical Sciences, Beijing 102206, China; 3International Academy of Phronesis Medicine, Guangzhou 510005, China

**Keywords:** hepatocellular carcinoma (HCC), RNF149, DNAJC25, ubiquitination, tumor microenvironment (TME)

## Abstract

**Simple Summary:**

Hepatocellular carcinoma (HCC) accounts for over 80% of cases among liver cancer, with high incidence and poor prognosis. Thus, it is of valuable clinical significance for discovery of potential biomarkers and drug targets for HCC. In this study, based on proteomics data analysis, really interesting new gene (RING) finger protein 149 (RNF149) was found to be elevated in hepatocellular carcinoma (HCC) tissues and associated with HCC malignancy, which was validated by immunohistochemistry (IHC) staining. RNF149 was demonstrated to promote proliferation, migration, and invasion of HCC cells dependent on its E3 ubiquitin ligase activity in vitro, and further bioinformatics analysis indicated that high expression of RNF149 correlated with immunosuppressive tumor microenvironment (TME). These results suggest that RNF149 could exert protumor functions in HCC in dependence of its E3 ubiquitin ligase activity, and might be a potential prognostic marker and therapeutic target for HCC treatment.

**Abstract:**

Hepatocellular carcinoma (HCC) accounts for over 80% of cases among liver cancer, with high incidence and poor prognosis. Thus, it is of valuable clinical significance for discovery of potential biomarkers and drug targets for HCC. In this study, based on the proteomic profiling data of paired early-stage HCC samples, we found that RNF149 was strikingly upregulated in tumor tissues and correlated with poor prognosis in HCC patients, which was further validated by IHC staining experiments of an independent HCC cohort. Consistently, overexpression of RNF149 significantly promoted cell proliferation, migration, and invasion of HCC cells. We further proved that RNF149 stimulated HCC progression via its E3 ubiquitin ligase activity, and identified DNAJC25 as its new substrate. In addition, bioinformatics analysis showed that high expression of RNF149 was correlated with immunosuppressive tumor microenvironment (TME), indicating its potential role in immune regulation of HCC. These results suggest that RNF149 could exert protumor functions in HCC in dependence of its E3 ubiquitin ligase activity, and might be a potential prognostic marker and therapeutic target for HCC treatment.

## 1. Introduction

Liver cancer ranks sixth in cancer incidence and third in cancer mortality worldwide [1], which remains a major health challenge. Among liver cancer, hepatocellular carcinoma (HCC) accounts for more than 80% of cases. While hepatitis B virus (HBV) infection is the main risk factor for HCC in Asia, nonalcoholic steatohepatitis (NASH) is becoming the fastest growing etiology of HCC, especially in the west [2]. Despite surgical resection at early stage, the recurrence rates reach 70% at five years [3]. The median overall survival of advanced-HCC patients with first-line systemic therapies is less than 20 months [2]. Thus, clarification of molecular mechanisms behind HCC initiation and progression, as well as discovery of potential prognostic markers and drug targets, are urgently needed.

With the rapid advancement of mass spectrometric techniques and bioinformatics, MS-based proteomics has provided deep insights into the characterization of HCC. For example, our previous study based on proteomics data from 101 pairs of early-stage HCC samples stratified HCC samples into three subtypes (S-I, S-II, and S-III) with increased malignancy, of which S-III subtype was associated with the poorest outcomes and featured cholesterol metabolism dysregulation [4]. Due to the heterogeneity and complexity of HCC, more in-depth analysis of proteomics data is also required.

A growing body of evidence revealed that dysregulation of E3 ubiquitin ligases is related to the occurrence, progression, and prognosis of various cancers, including HCC [5]. Among E3 ubiquitin ligases, really interesting new gene (RING)-finger (RNF) family is the largest one, with more than 300 validated members in humans, and mediates ubiquitination through their RING-finger domain [6]. RNF proteins have been reported to regulate crucial cellular processes, including cell cycle, DNA repair, cell signaling, and responses to hypoxia [7]. Several RNF proteins were found dysregulated in HCC, such as TRIM24 [8] and RNF181 [9]. TRIM24 expression correlates with poor prognosis, and promotes HCC progression through negatively regulating AMPK expression [8]. Instead, RNF181 is downregulated in HCC tissues and suppresses HCC growth by inhibiting the ERK/MAPK pathway [9]. However, whether there is any new player of the RNF family in HCC remains to be further investigated.

E3 ubiquitin protein ligase RING finger protein 149 (RNF149) belongs to the RNF family, and its gene is located on chromosome 2 in humans. Human RNF149 gene encodes a 400-amino-acid protein with a putative transmembrane domain in the middle region, and a RING-finger domain near its C terminus. It has been previously reported that *RNF149* gene is mutated in some human cancers, including breast, ovarian, and colorectal cancer. In colorectal cancer, RNF149 selectively degrades wild-type B-RAF instead of mutant B-RAF [10]. In nasopharyngeal carcinoma (NPC), MFSD4A-recruited RNF149 mediates the ubiquitination and degradation of EPHA2, thus inhibiting the proliferation, migration, and invasion of NPC cells [11]. Based on TCGA data, *RNF149* expression was also significant in many other cancers, including HCC and breast cancer (Appendix A). Nevertheless, the role of RNF149 in HCC is largely unknown.

In this study, based on our previous proteomics data, we found that RNF149 was upregulated in tumor tissues and correlated with poor diagnosis of HCC patients. Furthermore, we validated for the first time that RNF149 promoted HCC progression in vitro through its ubiquitin ligase activity, and identified DNAJC25 as its novel substrate. In addition, our bioinformatics analysis, for the first time ever, revealed that RNF149 expression was associated with immunocyte infiltration and T cell functions, indicating its potential role in immune regulation of HCC.

## 2. Materials and Methods

### 2.1. HCC Proteomics and Transcriptomics Data Analysis

HCC proteomics data were downloaded from the Proteomics Identification Database (https://www.ebi.ac.uk/pride/, accession number PXD006512, accessed on 5 September 2020) and were searched against the human UniProt database (version 20140922, 20,193 sequences) by MaxQuant (version 1.5.3.30, Max Planck Institute of Biochemistry, Munich, Germany) [4,12]. The Cancer Genome Atlas (TCGA)-Liver hepatocellular carcinoma (LIHC) transcriptomics data were downloaded from the Genomic Data Commons Data Portal (https://portal.gdc.cancer.gov/, accessed on 9 December 2020) [13].

Proteomics and transcriptomics data analysis was performed by R (version 4.2.2). Using R/Bioconductor package limma (version 3.54.0) to identify the differentially expressed proteins between HCC tissues and peritumor tissues, and proteins that meet the following criteria are considered as tumor upregulated proteins: (1) fold-change larger than 2; (2) *p* less than 0.05. R/Bioconductor package enrichR (version 3.2), clusterProfiler (version 4.6.0), Gene Set Variation Analysis (version 1.46.0) was used for over-representation enrichment analysis (ORA), Gene Set Enrichment Analysis (GSEA), single sample GSEA (ssGSEA,), respectively [14,15,16,17]. R packages survival (version 3.5–0) and survminer (version 0.4.9) were utilized to perform Cox regression and Kaplan–Meier analysis on patients’ overall survival (OS) and disease-free survival (DFS), and provide a comparison between various groups. Pearson correlation was used for correlation analysis. Wilcoxon rank-sum test was used to compare two groups.

Based on the immune-related signature gene list [18,19,20], ssGSEA was performed to assess the immune enrichment score and the immune cell infiltration, such as activated CD8+ T cell, exhausted T cell, M2 macrophage, etc. Enrichment score heatmap was plotted using the R/Bioconductor package ComplexHeatmap (v2.4.3). Pearson correlation was used to calculate correlations between RNF149 protein expression and immune cell infiltration.

### 2.2. Cell Culture

The human HCC cell lines HepG2 (Serial: SCSP-510), Huh7 (Serial: SCSP-526), Hep3B (Serial: SCSP-5045), PLC/PRF/5 (Serial: SCSP-5095), and SNU-387 (Serial: SCSP-5046), and the human embryonic kidney cell line HEK293T (Serial: SCSP-502) were obtained from Cell Bank/Stem Cell Bank, Chinese Academy of Sciences. The human HCC cell line SNU-475 cell line was purchased from Shanghai Zhong Qiao Xin Zhou Biotechnology. The human HCC cell line MHCC-97H cell line was obtained from Dr. Ying Jiang (National Center for Protein Sciences).

The HepG2, Huh7, Hep3B, PLC/PRF/5, MHCC-97H, SNU-387, SNU-475, and HEK293T cell lines were cultured in either Dulbecco’s modified Eagle’s medium (DMEM, Gibco, Grand Island, NY, USA) or Roswell Park Memorial Institute 1640 medium (RPMI 1640, Gibco, Grand Island, NY, USA) supplemented with 10% fetal bovine serum (FBS, Newzerum, Christchurch, New Zealand) and 1% Pen Strep (Gibco, Grand Island, NY, USA) at the conditions of 37 °C and 5% CO_2_. All of the cells were confirmed by short tandem repeat (STR) assay and contained no mycoplasma contamination.

### 2.3. Cell Transfection and Stable Cell Line Generation

For siRNA transfection, two different siRNAs targeted for RNF149 were purchased from JTS scientific (Wuhan, China), and sequences are as follows: siRNF149-1 *5′-CCAGATGATGACGGAAGTGTT-3′*, siRNF149-2 *5′-GAATTGATGTTGATGCTGATT-3′*. The siRNAs were transfected into cells using TurboFect reagent (Thermo Fisher Scientific, Waltham, MA, USA) according to its manual. A volume of 2 μg siRNA (or control) was added to 100 μL serum-free medium and incubated with 7 μL TurboFect for 20 min. Cells with a density of about 50% in 6-well plates were transfected, and replenished with fresh DMEM after 12 h of incubation.

For plasmid transfection, RNF149 cDNA was cloned into pCMV-HA and pHAGE-3 × FLAG vectors, and the plasmids were transfected into cells using TurboFect reagent (Thermo Fisher Scientific, Waltham, MA, USA) according to its manual.

Stable cell lines with RNF149 overexpression were constructed using pHAGE-3 × FLAG vector. Volumes of 12 μg of pHAGE-3 × FLAG vector or pHAGE-3 × FLAG-RNF149 plasmid, 3 μg of pMD2.G plasmid, and 9 μg of psPAX2 plasmid were added to 500 μL serum-free medium, and incubated with polyetherimide (23966-1, Polyscience, Warrington, PA, USA) for 20 min. HEK293T cells with a density of about 70% in 10 cm dishes were transfected. After 12 h of incubation, cells were replenished with fresh DMEM. The culture medium was collected at 48 h and 72 h after transfection, and the medium was filtered through a 0.45 μm filter membrane to obtain a lentiviral virus solution. The cells with a density of about 50% in the 6-well plate were infected with the concentrated virus solution and 4 μg/mL polybrene (Sigma-Aldrich, Darmstadt, Germany). After incubation for 12 h, the medium was replaced with fresh DMEM. After 72 h of infection, 2 μg/mL puromycin (Thermo Fisher Scientific, Waltham, MA, USA) was added to the medium to screen for the stable RNF149-overexpressing cells.

### 2.4. Antibodies

Antibody against Flag tag (A8592) was purchased from Sigma-Aldrich (Darmstadt, Germany). Antibody against β-Actin (66009-1-IG) was purchased from Proteintech (Wuhan, China). Antibody against RNF149 (HPA011424) was purchased from Atlas Antibodies (Bromma, Sweden). Horseradish peroxidase-labeled goat anti-mouse (SA00001-1) and goat anti-rabbit (SA00001-2) immunoglobulin G (IgG) (H + L) antibodies were purchased from Proteintech (Wuhan, China).

### 2.5. Western Blotting

Total cellular proteins were extracted using NETN lysis buffer (10 mM Tris, pH 8.0, 100 mM NaCl, 0.5% NP-40 and 1 mM EDTA) supplemented with protease inhibitor phenylmethylsulfonyl fluoride (PMSF) (Solarbio life sciences, Beijing, China) and phosphatase inhibitor NaF. After blowing and mixing, the cells were put on ice for 10 min and sonicated for 5 min. Protein concentration was quantified using BCA Protein Assay Kit (23227, Thermo Fisher Scientific, Waltham, MA, USA). The lysis product was mixed with the 5 × loading buffer (50 mM Tris, pH 6.8, 2% SDS, 10% glycerol, 0.005% bromophenol blue and 2% β-mercaptoethanol) in a 4:1 ratio, and the sample was boiled twice at 95 °C for 7 min each time to obtain the WB sample.

Western blot analysis followed standard procedures [21]. The sample was loaded on a 10% SDS-PAGE gel and subjected to 120 V constant voltage electrophoresis until bromophenol blue ran out of the gel. The separated sample was electrolyzed at 250 mA for 90 min to a 0.45 nitrocellulose filter membrane (66485, BioTrace, Auckland, New Zealand). Nitrocellulose membrane was incubated with 5% skimmed milk dissolved in TBST at room temperature for 2 h. After blocking, the primary antibody was added and incubated overnight at 4 °C, and then washed three times with TBST, 7 min each time. The corresponding secondary antibody was added at room temperature and incubated for 1 h. Finally, the membrane was washed three times with TBST for 7 min each time. The target protein was visualized using NcmECL Ultra (P10200, NCM Biotech, Suzhou, China).

### 2.6. Proliferation and Colony Formation Assays

For the proliferation assay, 5000 cells were cultured in 96-well cell culture plates for 6 days, and cell proliferation was detected using Cell Counting Kit-8 (AQ308-3000T, Beijing Aoqing Biotechnology, Beijing, China) according to the manual, once each day.

For the colony formation assay, 4000 cells were cultured in 6-well cell culture plates for 14 days, and then fixed in methanol for 30 min and stained with crystal violet solution (C0121, Beyotime Biotechnology, Beijing, China) for 1 h. Representative images were acquired using a camera. The colony numbers were counted by Photoshop CS, and the data were analyzed using GraphPad Prism 7 software.

### 2.7. Migration and Invasion Assays

For cell migration or invasion, transparent PET membrane with 8 μm pores (353097, BD Biosciences, Franklin Lakes, NJ, USA) or Corning Matrigel Invasion Chamber 24-Well Plate 8.0 Micron (354480, BioCoat, Horsham, PA, USA) were used. Huh7 (1 × 10^5^), MHCC-97H (3 × 10^5^), PLC/PRF/5 (3 × 10^5^) or SNU-475 (1 × 10^5^) cells were seeded in the upper chamber with 300 μL serum-free medium, and 500 μL medium containing 10% FBS was added to the lower chamber. The incubation time for migration or invasion of Huh7 and SNU-475 was 24 h, and the incubation time for migration or invasion of MHCC-97H and PLC/PRF/5 was 60 h. After incubation, the cells were fixed in methanol and stained with crystal violet solution (C0121, Beyotime Biotechnology, Beijing, China). Representative images were acquired using an optical microscope (Nikon, Tokyo, Japan), and all images were taken from a random field of view. The cell numbers were counted by Photoshop CS and the data were analyzed using GraphPad Prism 7 software.

### 2.8. Immunohistochemistry (IHC)

A tissue microarray (TMA) chip containing 94 HCC and 86 paired nontumor liver tissues were subjected to immunohistochemical staining at Shanghai Outdo Biotech Company using antibody against RNF149 (HPA011424, Atlas Antibodies, Bromma, Sweden). The chip’s protein intensity was scored by two trained pathologists, blindly and independently. The staining extent score ranged from 0 to 4 based on the percentage of immunoreactive tumor cells (0%, 1–5%, 6–25%, 26–75%, and 76–100%). The staining intensity was scored as negative (score = 0), weak positive (score = 1), positive (score = 2), or strong positive (score = 3). A score ranging from 0 to 12 was calculated by multiplying the staining extent score by the intensity score.

### 2.9. Co-Immunoprecipitation (Co-IP)

Total cellular proteins were extracted using NETN lysis buffer freshly supplemented with protease inhibitors (A32955, Thermo Scientific, Waltham, MA, USA) and phosphatase inhibitors (04906837001, Roche, Basel, Switzerland). After the whole cell lysate was obtained according to the previous method and the protein concentration was determined, 55 μL of the lysate was taken as the input group, mixed with the loading buffer, and boiled to obtain the WB sample.

For the protein with Flag tag, 30 μL Flag-agarose beads (A2220, Sigma-Aldrich, Darmstadt, Germany) were added to the remaining lysis products, and slowly mixed at 4 °C for 6 h. The agarose beads were washed with NETN lysis buffer 3 times, and slowly mixed at 4 °C for 5 min each time. The supernatant was discarded, and 55 μL 2 × loading buffer was added to the agarose beads. The sample was boiled to obtain the IP sample.

For RNF149, 20 μL Protein A/G PLUS agarose beads (sc-2003, Santa Cruz, CA, USA) and 1 μg Rabbit IgG (3900S, Cell Signaling Technology, Danvers, MA, USA) were added to the remaining lysates, and slowly mixed at 4 °C for 2 h to remove nonspecific binding proteins. The supernatant was taken, and 2 μg Rabbit IgG or RNF149 antibody was added to the control group or the experimental group, respectively, and slowly mixed at 4 °C overnight. The IP sample was obtained by adding 20 μL Protein A/G PLUS agarose beads, slowly mixing at 4 °C for 4 h, and adding 2 × loading buffer to boil the sample. The enriched proteins were then detected by WB.

### 2.10. IP-MS

Huh7 cells with RNF149 overexpression were treated with 50 μM MG132 (S2619, selleck, Houston, TX, USA) for 6 h, and then were harvested for preparation of whole cell lysates. After incubation with anti-Flag-agarose beads, the protein was subjected to SDS-PAGE. After electrophoresis, the protein gel was stained with Coomassie brilliant blue, and then analyzed by MS by QE-HF for 75 min.

### 2.11. Proteomic Profiling of HCC Cells

Huh7 cells with RNF149 overexpression and MHCC-97H cells with RNF149 knockdown were extracted by SDC lysis buffer (100 mM Tris, pH 8.5, 1% DOC, 10 mM TCEP, 40 mM CAA). The protein concentration was determined using the BCA Protein Assay Kit. Proteins were digested by trypsin (protein:enzyme, 50:1, *w*/*w*) overnight. The digested tryptic peptides were acidified by FA to a final FA concentration of 1%, and then centrifuged at 16,000× *g* for 10 min. The acidified supernatants were desalted using a homemade C18 stage-tip. The homemade C18 stage-tip was activated with acetonitrile and then equilibrated twice with 0.1% formic acid. The acidified supernatant was loaded to the C18 stage-tip, and then 0.1% formic acid was used to wash the column twice, and 50% acetonitrile with 0.1% formic acid was used to elute peptides. The peptides were analyzed by MS by QE-HF for 150 min.

### 2.12. Xenograft Tumor Growth Assays

Male NOD-SCID mice (SpePharm Biotechnology, Beijing, China), aged 4 weeks, were housed in a special pathogen-free animal facility. Each mouse was injected subcutaneously with 4 × 10^6^ RNF149 OE or empty vector (EV) control Huh7 cells into the right or left flank. The tumor length and short diameter were measured every three days with vernier calipers until 30 days later. Mice were euthanized and tumors were retained for follow-up analysis. Tumor volumes were determined using the ellipsoidal volume formula: 1/2 × length × width^2^. All animal experiments were performed according to our Institutional Animal Care and Use Committee (IACUC) guidelines.

### 2.13. Statistical Analysis

All data are recorded as mean ± SEM. Statistical analyses were performed with GraphPad Prism software (Version 8.0) (San Diego, TX, USA) using unpaired two-tailed Student’s *t*-test, as described in the figure legends. *p* value less than 0.05 was considered statistically significant.

## 3. Results

### 3.1. RNF149 Expression Is Highly Elevated in HCC Tissues and Correlated with Poor Prognosis

To identify novel RNF proteins involved in HCC progression based on our previous proteomics data [4], we combined 1474 proteins upregulated in HCC tissues compared with paired paracancerous tissues (Log_2_FC > 2, *p* < 0.01) and 90 RNF proteins detected in HCC tissues, which led to 17 protein candidates (Figure 1A,B). Consistently, most candidates present high hazard ratio (HR) values in overall survival (OS) and disease-free survival (DFS), implying their correlations with unfavorable prognosis (Figure 1B). As TRIM24 has been reported to promote the progression of HCC [8], we therefore focused on RNF149 (FC = 16.91, OS HR = 4.27, DFS HR = 3.18), which was second only to TRIM24 and with no report in HCC before.

According to our data, RNF149 was significantly upregulated in tumor tissues (Figure 1C), and its expression increased profoundly along with the rising malignancy of three subtypes of HCC [4] (Figure 1D). In line with that, expression of RNF149 showed significant upregulation in HCC patients with high serum alpha-fetoprotein (AFP) levels and positively correlated with AFP levels (Figure 1E,F), further confirming the correlation of RNF149 with HCC malignancy. Notably, the OS and DFS in RNF149 high-expression group were significantly shorter than that in RNF149 low-expression group (Figure 1G), suggesting that RNF149 was a prognostic marker with poor outcomes.

In addition, the HCC cohort we used produced matched transcriptomics data, and Pearson correlation analysis showed that *RNF149* mRNA expression level was highly consistent with its protein level (correlation coefficient = 0.4, *p* = 2.5 × 10^−7^) (Appendix A). We further analyzed the mRNA expression of *RNF149* in the same HCC cohort. Consistent with the protein level results, *RNF149* mRNA was significantly upregulated in tumor tissues and highly expressed in S-III subtypes of HCC (Appendix A). Kaplan–Meier survival analysis also showed that high expression of *RNF149* mRNA was associated with shorter OS and DFS (Appendix A). The mRNA expression of *RNF149* also showed significant upregulation in HCC patients with high serum AFP levels and positively correlated with AFP levels (Appendix A). Because the degree of differentiation was not provided in the cohort, we only analyzed the correlation between *RNF149* protein and mRNA expression and MVI, metastasis, TP53 mutation, and MYC amplification. As shown in Appendix A, *RNF149* was not significantly associated with those prognostic markers.

Furthermore, we investigated *RNF149* expression using data in the TCGA database. Consistent with our proteomics results, RNF149 expression showed striking elevation in HCC tissues (Figure 1H) and was associated with shorter OS and DFS (Figure 1I), which further confirmed that *RNF149* was notably upregulated in HCC tissues and high expression of *RNF149* indicated unfavorable prognosis.

### 3.2. IHC Staining Experiments Validates That RNF149 Is an Unfavorable Prognostic Marker

To verify our results from omics data, we conducted IHC staining experiments to determine the RNF149 expression in an independent HCC cohort using the tumor tissue microarray. As shown in Figure 2A–C, HCC tissues presented stronger positive staining of RNF149 compared with paired paracancerous tissues, indicating the significant elevation of RNF149 in HCC tissues. Moreover, a high level of RNF149 was indicative of shorter OS (Figure 2D). Consistently, RNF149 expression was strikingly higher in HCC samples with high pathological grade (III, IV) than that with low pathological grade (I, II) (Figure 2E). Thus, these results validated that RNF149 is an unfavorable prognostic marker.

### 3.3. Molecular Characteristics of RNF149-High HCC Reveal Enhanced Malignancy

To characterize the molecular features related to RNF149 expression, HCC samples [4] were divided into RNF149-high group and RNF149-low groups according to the RNF149 abundance, and gene ontology (GO) enrichment analysis was performed on the upregulated proteins in each group, respectively. The results showed that the RNF149-high group mainly enriched cell cycle, mRNA processing, and focal adhesion pathways (Figure 3A,B). Gene set enrichment analysis (GSEA) further revealed that proliferation, ribosome biogenesis, metastasis, and epithelial mesenchymal transition were enriched in the RNF149-high group (Figure 3C), suggesting that the high expression of RNF149 was related to the activation of proliferation, migration, and invasion of HCC.

To further verify our results, we constructed Huh7 RNF149-overexpressing and MHCC-97H RNF149-knockdown cells with high or low basal expression of RNF149, respectively (Appendix A), and profiled the differentially expressed proteins of those cells based on MS. In line with our expectations, proteins associated with pathways related to cell cycle and migration were significantly upregulated in RNF149-overexpressing cells and downregulated in RNF149-knockdown cells (Figure 3D). Therefore, we supposed that RNF149 might promote the development of HCC by activating proliferation, migration, and invasion.

### 3.4. RNF149 Promotes the Proliferation, Migration, and Invasion of HCC Cells

Consistent with our assumptions, overexpression of RNF149 in Huh7 cells significantly promoted cell proliferation, as well as cell colony formation (Figure 4A,C and Appendix A), and knockdown of RNF149 in MHCC-97H cells and PLC cells profoundly inhibited cell proliferation as well as cell colony formation (Figure 4B,D and Appendix A). To investigate whether the change of RNF149 protein level affects the sensitivity of standard-of-care chemotherapeutics, the IC_50_ of sorafenib after overexpression or knockdown of RNF149 in Huh7 or MHCC-97H cells was detected. We found that the IC_50_ of sorafenib increased after overexpression of RNF149, and decreased after knockdown of RNF149 (Figure 4E). Moreover, transwell assays demonstrated that overexpression or knockdown of RNF149 also induced or reduced cell migration and invasion capacity of HCC cells, respectively (Figure 4F–H). We examined the changes of EMT markers after knocking down RNF149 in MHCC-97H and PLC cells, respectively. Our results showed that after knocking down RNF149, N-Cadherin was significantly downregulated in MHCC-97H cells and Vimentin was significantly downregulated in PLC cells (Appendix A), indicating that RNF149 might promote EMT, which is consistent with our hypothesis. Furthermore, in the subcutaneous tumor formation assay, RNF149-overexpressing Huh7 cells exhibited a significantly increased tumor size and weight of cells compared with the control group (Figure 5A–C). These results validated that RNF149 functions as an oncoprotein by stimulating cell proliferation, migration, and invasion.

### 3.5. RNF149 Promotes the Progression of HCC through Its E3 Ubiquitin Ligase Activity

As RNF149 is an E3 ubiquitin ligase, we further investigated whether its ligase activity was required for its tumor-promoting functions. As shown in Figure 6A, we constructed cells with overexpression of RNF149 wild-type (WT) or H289A mutant, which lacks E3 ubiquitin ligase activity [10]. Notably, while overexpression of RNF149 WT promoted HCC cell growth significantly, overexpression of RNF149 H289A mutant showed no effect on proliferation (Figure 6B). Similar results were also observed in colony formation assays (Figure 6C). Moreover, unlike RNF149 WT, overexpression of RNF149 H289A mutant did not stimulate cell migration and invasion in vitro (Figure 6D,E). Hence, our results demonstrated that RNF149 promoted HCC progression in an E3 ubiquitin ligase activity-dependent manner.

### 3.6. Proteomics Data Identifies DNAJC25 as a Potential Substrate of RNF149 in HCC Cells

To identify the potential substrate of RNF149 in HCC cells, we supposed that it should be downregulated strikingly in RNF149-overexpressing cells and interact with RNF149. With the combination of 250 significantly-decreased proteins (Log_2_FC > 2, *p* < 0.05) in RNF149-overexpressing cells using proteomics methods and 54 RNF149-interacting proteins (Log_2_FC > 2, *p* < 0.01) from IP-MS experiments (Figure 7A,B), DnaJ homolog subfamily C member 25 (DNAJC25) [22] was shown to be the candidate protein (Figure 7C). Consistently, DNAJC25 showed significant negative correlation with RNF149 and profound downregulation in HCC tissues based on our previous data from 101 pairs of HCC samples [4] (Figure 7D,E). Furthermore, DNAJC25 expression was also negatively associated with increased malignancy of HCC (Figure 7F), and high expression of DNAJC25 in tumor tissues indicated longer OS and DFS (Figure 7G). These results demonstrated that DNAJC25 might be a potential substrate of RNF149 in HCC cells.

### 3.7. RNF149 Interacts with DNAJC25 and Promotes its Degradation

Through Co-IP experiment, we validated the interaction of DNAJC25 and RNF149 in cells (Figure 7H). Moreover, overexpression of RNF149 in Huh7 cells significantly reduced DNAJC25 protein levels, while overexpression of RNF149 H289A mutant did not (Figure 7I), implying the requirement of E3 ubiquitin ligase activity. Notably, MG132 treatment rescued DNAJC25 protein levels in RNF149-overexpressing cells (Figure 7J), which indicated that RNF149 promoted ubiquitin proteasome system (UPS)-dependent degradation of DNAJC25. Therefore, these results demonstrated that DNAJC25 was a novel target of RNF149 in HCC.

### 3.8. High RNF149 Expression Correlates with Immunosuppressive Tumor Microenvironment (TME)

Immune checkpoint inhibitors (ICIs) have been approved for HCC treatments. However, only 7–20% HCC patients can benefit from immunotherapy [23]. The tumor immune microenvironment (TIME) plays a significant role in response to immunotherapy [24,25], and according to the previous analysis results, the RNF149-high group had significantly enriched immune response pathways (Figure 2A and Figure 8A). Hence, we further explored the relationship between RNF149 and HCC immune microenvironment. The infiltration abundance of different immune cells [18,19,20] in HCC patients was characterized. The results suggest that the overall immune enrichment scores were higher in thee RNF149-high group characterized with higher infiltration of immunosuppressive cells, such as M2 macrophages, myeloid-derived suppressor cells (MDSCs), regulatory T cells, and exhausted T cells (Figure 8B–D). Intriguingly, there was no significant difference in antitumor immune cells, such as M1 macrophages, activated CD8 + T cells, and natural killer cells between the RNF149-high and -low groups (Figure 8C). Thus, our results supported the notion that high RNF149 levels were associated with immunosuppressive tumor microenvironment.

## 4. Discussion

As a major health concern globally, research on HCC has attracted much attention. With the aid of advanced proteomics technology, researchers have been able to profile the proteomic landscape of HCC tissues from clinical cohorts, and have revealed multiple molecular aspects associated with HCC progression, treatment, and outcome. Our previous study identified sterol O-acyltransferase 1 (SOAT1) as a prognostic marker with unfavorable outcome and potential drug target for treatment of S-III subtype with highest malignancy [4,26]. In addition, another cohort study based on proteomics data discovered that phosphorylation of fructose-bisphosphate aldolase A (ALDOA) promoted HCC progression by enhancing glycolysis [27]. Thus, it is conceivable that deep analysis of proteomics data will gain us a more insightful and comprehensive understanding towards the critical molecular mechanisms behind HCC, contributing to the identification of potential drug targets and biomarkers.

Through analyzing our previous proteomics data, we found 17 members of the RNF family with significant elevation in tumor tissues. Consistently, some RNF proteins among them have been reported to be related with HCC. For example, TRIM24 overexpression is related to HCC onset and progression, and functions as a tumor promoter by downregulating AMPK levels [8]. HLTF expression is upregulated in HCC tissues, and overexpression of HLTF accelerates growth and metastasis of HCC cells via stabilizing SRSF1 and stimulating ERK/MAPK pathway [28]. Notably, we identified RNF149 as a novel player of RNF proteins in HCC, which showed significant upregulation, preceded only by TRIM24. RNF149 expression positively correlated with AFP level and poor prognosis. Moreover, immunohistochemical experiments of independent HCC cohorts further validated that RNF149 expression was upregulated in tumors, and could be a prognostic marker.

As an E3 ubiquitin ligase, only two experimentally validated substrates of RNF149 were reported so far: wild-type but not mutant BRAF in colorectal cancer [10], and EPHA2 in NPC [11]. To discover the substrate of RNF149 in HCC, we focused on the proteins which both showed significant downregulation in RNF149-overexpressed HCC cells and interacted with RNF149, and DNAJC25 was the only candidate with high confidence. Further biochemical experiments proved DNAJC25 was a substrate of RNF149. Consistently, DNAJC25 was reported be downregulated in HCC [22]. Intriguingly, B-RAF and EPHA2 were not included in our candidate list, because B-RAF did not interact with RNF149 based on our Co-IP experiments, and EPHA2 protein levels did not decrease profoundly when RNF149 was overexpressed, which might be resulting from differences of experimental conditions and cancer cell lines. More substrate proteins of RNF149 need to be determined in the future.

Due to a putative single-pass transmembrane domain in its middle region, RNF149 is believed to be localized in membranes, such as endoplasmic reticulum (ER) membrane. However, reports regarding its functions in cells are few. It has been reported that RNF149 promotes cell survival in colorectal cancer [10] and inhibits cell migration in NPC [11]. Additionally, RNF149 is found to involve the development of rat neonatal gonocytes [29]. Recently, a study showed that RNF149 interacts with components of pre-emptive ER associated quality control (pEQC) pathway, regulating the translocation of misfolded proteins [30]. In our research, we proved that RNF149 promoted cell growth and migration of HCC cells, and these tumor-promoting effects were dependent on its E3 ubiquitin ligase activity (Figure 9). Notably, DNAJC25 is predicted to be ER localized and associated with protein folding [22,31], indicating that RNF149 may regulate quality surveillance of ER proteins through DNAJC25. In addition, enrichment analysis of differentially expressed proteins in RNF149-overexpressed or RNF149-knockdown cells also suggested that RNF149 might play a role in many cellular processes, such as mitotic cell cycle, fatty acid, and pyruvate metabolism, which remains to be further investigated.

Antitumor immunotherapy has attracted increasing attention, and immune checkpoint inhibition with antivascular endothelial growth factor (VEGF) neutralizing antibodies has become first-line therapy for advanced-HCC patients [32]. Thus, we also explored the potential role of RNF149 in immune regulation of HCC. Based on proteomics data, we found that markers of CD4+ T cells and MDSCs were enriched in patients with high RNF149 expression. Additionally, RNF149 expression was correlated with T cell exhaustion and Treg cells. These results indicated an immunosuppressive feature of tumor microenvironment in RNF149-upregulated patients, and further study is needed to validate the functions of RNF149 in TME regulation.

## 5. Conclusions

In summary, our study not only demonstrated that RNF149 expression correlated with HCC malignant grades, and RNF149 triggered HCC progression through its ubiquitin ligase activity and thus degrading DNAJC25, but also indicated that RNF149 was associated with TME regulation in HCC. Therefore, RNF149 might be a potential prognostic marker and drug target for HCC treatment, and more efforts are required to clarify its cellular functions and molecular mechanisms in cancers for better clinical translation.

## Figures and Tables

**Figure 1 cancers-15-05203-f001:**
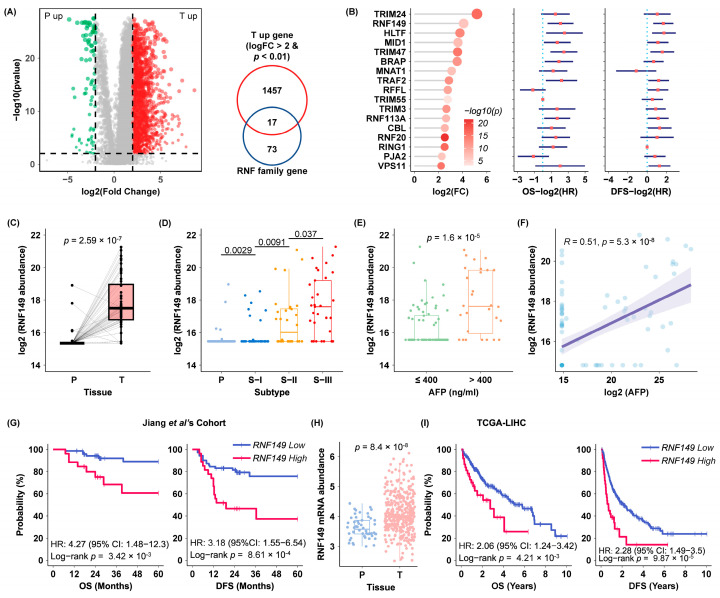
RNF149 expression is elevated in HCC tissues and correlated with poor prognosis of HCC patients. (**A**) Volcano plot of differentially expressed proteins in HCC tissues compared with noncancerous tissues (left), and Venn diagram of significantly upregulated proteins in tumor tissues and RNF proteins detected in HCC samples (right). (**B**) Dot plot of the 17 candidate proteins ranked by fold change (FC) with their hazard ratio (HR) values of overall survival and disease-free survival. (**C**) Expression of RNF149 protein in HCC tissues and paracancerous tissues. (**D**) RNF149 protein expression among three subtypes of HCC tissues. (**E**) RNF149 protein expression in AFP-low (≤400 ng/mL) and AFP-high (>400 ng/mL) groups. (**F**) Correlation analysis between the serum AFP levels and RNF149 protein expression. (**G**) Kaplan–Meier survival analysis for OS and DFS of HCC patients with different RNF149 protein expression under the optimal cut-off value [4]. (**H**) Expression of RNF149 mRNA in liver cancer tissues and paracancerous tissues. (**I**) Kaplan–Meier survival analysis for OS and DFS of HCC patients with different RNF149 mRNA expression under the optimal cut-off value.

**Figure 2 cancers-15-05203-f002:**
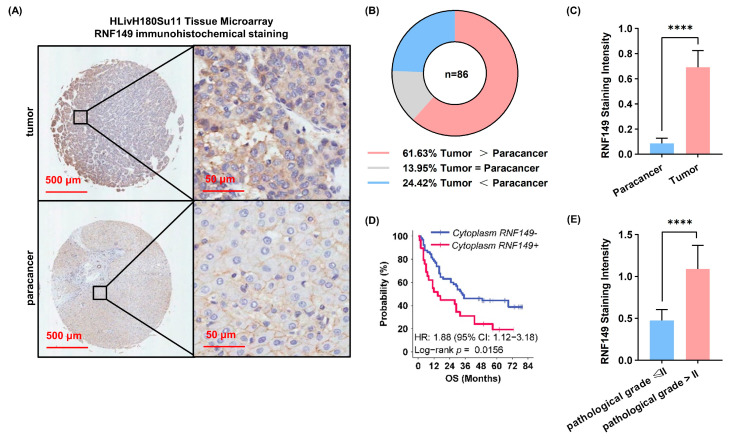
IHC staining verifies that RNF149 is an unfavorable prognostic marker. (**A**) Representative IHC staining images of HCC and paracancerous tissues. (**B**) Pie chart of overall RNF149 expression pattern between HCC and paracancerous tissues. (**C**) RNF149 IHC staining scores of RNF149 in cancer tissues and paracancerous tissues. (**D**) Kaplan–Meier survival analysis for OS of HCC patients with different IHC staining scores under the optimal cut-off value. (**E**) RNF149 staining scores in patients with low pathological grade (I, II) or high pathological grade (III, IV), respectively. **** *p* < 0.0001.

**Figure 3 cancers-15-05203-f003:**
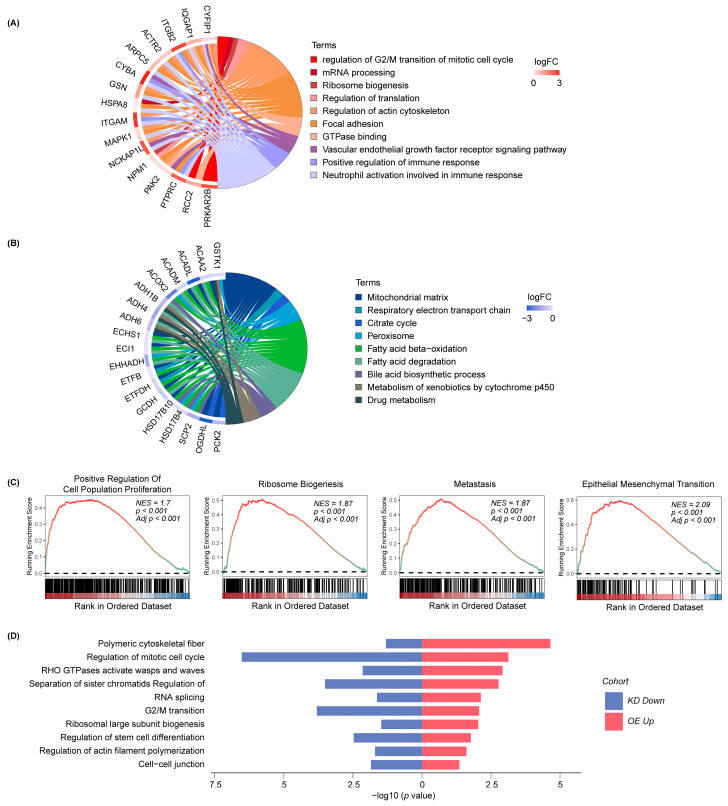
High expression of RNF149 is related to the activation of tumor proliferation, migration, and invasion-related pathways. (**A**,**B**) Chord diagrams of significantly upregulated proteins in HCC patients of high RNF149 levels tagged with enriched pathways (left), and significantly upregulated proteins in HCC patients of low RNF149 levels tagged with enriched pathways (right). (**C**) GSEA of proliferation and metastasis-related pathways in the HCC patients with high RNF149 levels. (**D**) GO analysis of notably elevated proteins in RNF149-overexpressing cells (red) and notably decreased proteins in RNF149-knockdown cells (blue).

**Figure 4 cancers-15-05203-f004:**
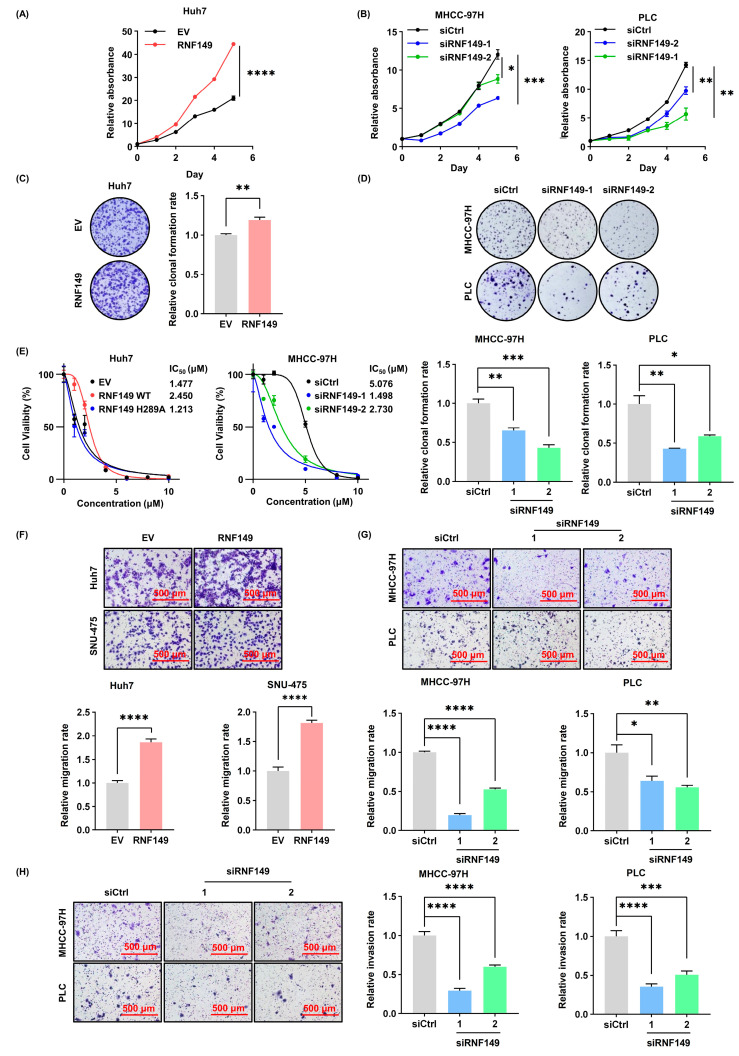
RNF149 promotes HCC cell proliferation, migration, and invasion. (**A**) Huh7 cells were transfected with indicated plasmids, and cell proliferation was determined using CCK-8 assays. (**B**) MHCC-97H and PLC cells were transfected with indicated siRNAs, and cell proliferation was determined using CCK-8 assays. (**C**,**D**) HCC cells were transfected with indicated plasmids or siRNAs. Cell proliferation was determined using colony formation determined by crystal violet staining. (**E**) HCC cells were transfected with indicated plasmids or siRNAs, and the IC_50_ of sorafenib was detected. (**F**–**H**) HCC cells were transfected with indicated plasmids or siRNAs. Cell migration (**F**,**G**) and invasion (**H**) were determined by transwell assays. * *p* < 0.05, ** *p* < 0.01, *** *p* < 0.001, **** *p* < 0.0001.

**Figure 5 cancers-15-05203-f005:**
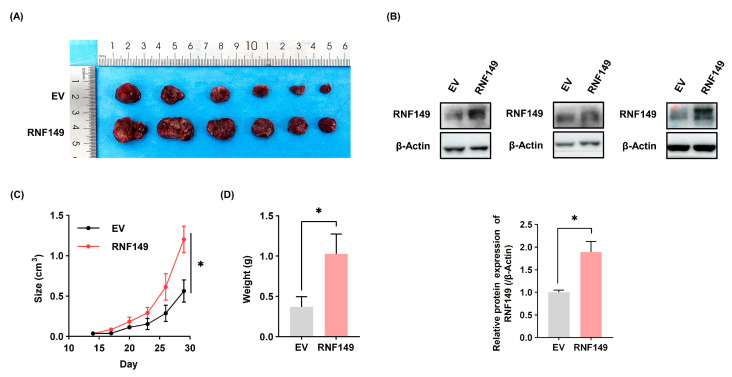
RNF149 promotes subcutaneous tumorigenesis in mice. (**A**) Subcutaneous tumor formation assays were performed in NOD-SCID mice using Huh7 cell lines stably overexpressing RNF149. Representative pictures are shown. (**B**) Protein expression of RNF149 in the recovered tumors was detected by WB, and the representative images of No. 1, No. 4, and No. 6 mice are displayed. (**C**,**D**) Tumor size and weight were determined (*n* = 6). * *p* < 0.05. The uncropped bolts are shown in Appendix A.

**Figure 6 cancers-15-05203-f006:**
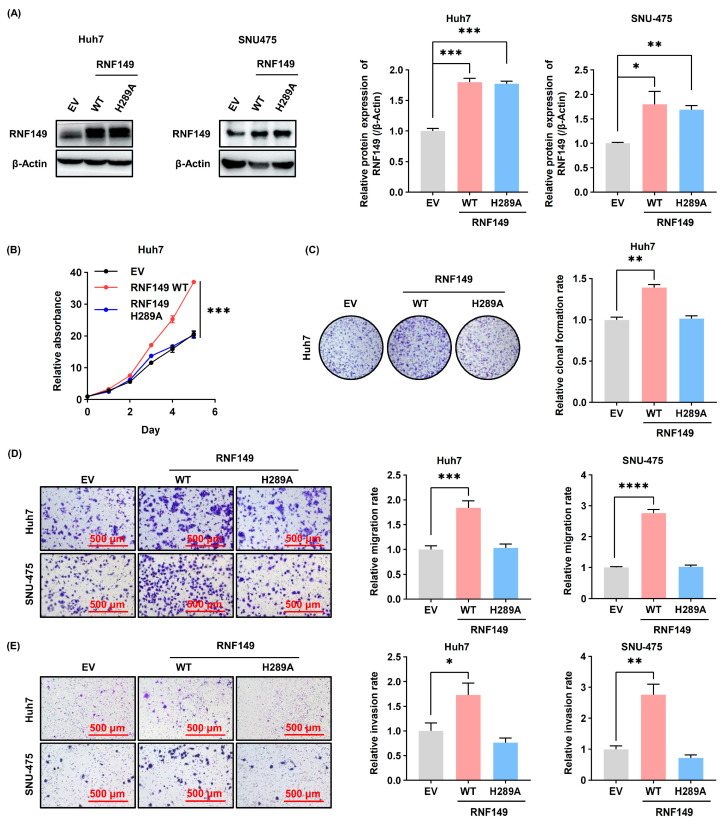
E3 ubiquitin ligase activity is required for RNF149-mediated progression of HCC in vitro. (**A**) Huh7 and SNU-475 cells were transfected with indicated plasmids, and RNF149 levels were detected by WB. (**B**,**C**) HCC cells were transfected with indicated plasmids. Cell proliferation was determined using CCK-8 assays (**B**), and colony formation was determined by crystal violet staining (**C**). (**D**,**E**) HCC cells were transfected with indicated plasmids, and cell migration (**D**) as well as invasion (**E**) were determined by transwell assays. * *p* < 0.05, ** *p* < 0.01, *** *p* < 0.001, **** *p* < 0.0001. The uncropped bolts are shown in Appendix A.

**Figure 7 cancers-15-05203-f007:**
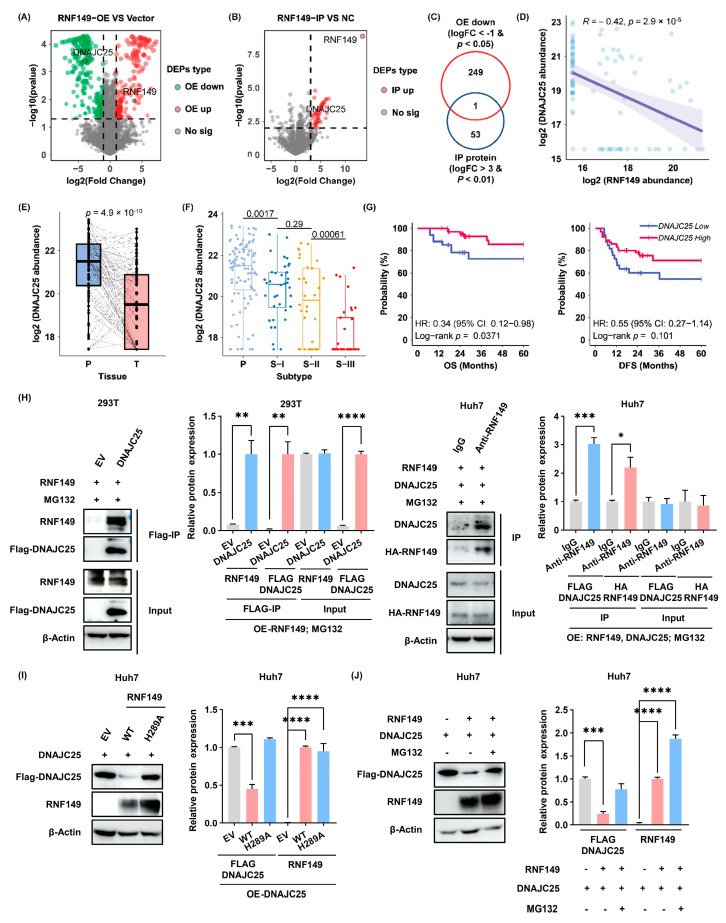
Proteomics data identifies DNAJC25 as a potential substrate of RNF149 in HCC cells, and high expression DNAJC25 was associated with the good prognosis of HCC patients. (**A**) Volcano plot of upregulated proteins (red dots) and downregulated proteins (green dots) in RNF149-overexpressing cells compared with control cells. (**B**) Volcano plot of significantly upregulated proteins in RNF149-IP samples compared with negative control samples. (**C**) Venn diagram of significantly elevated proteins in RNF149-overexpressing cells and RNF149-IP samples. (**D**) Correlation analysis of the protein expression between RNF149 and DNAJC25 in our previous proteomics data. (**E**) DNAJC25 protein expression in HCC tissues and noncancer tissues. (**F**) Comparison of DNAJC25 protein expression among three subtypes of HCC tissues. (**G**) Kaplan–Meier survival analysis for OS and DFS of HCC patients with different DNAJC25 protein expression under the optimal cut-off value. (**H**) Cells were transfected with indicated plasmids and the interaction of RNF149 and DNAJC25 was detected using IP and WB experiments. (**I**,**J**) Huh7 cells were treated with indicated treatment, and the effect of RNF149 overexpression (**I**) and MG132 incubation (**J**) on DNAJC25 protein levels was detected by WB. * *p* < 0.05, ** *p* < 0.01, *** *p* < 0.001, **** *p* < 0.0001. The uncropped bolts are shown in Appendix A.

**Figure 8 cancers-15-05203-f008:**
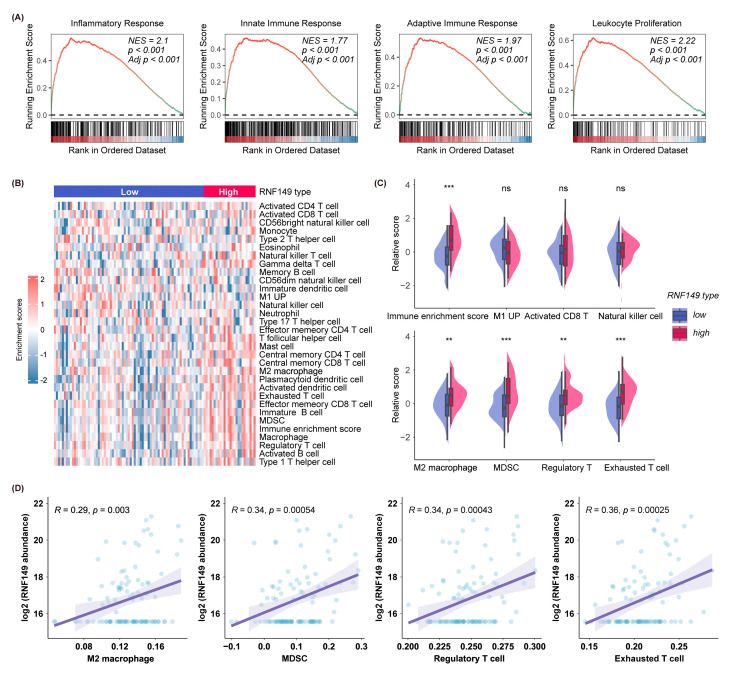
High RNF149 levels correlate with immunosuppressive tumor microenvironment (TME). (**A**) GSEA of immune response related pathways in HCC patients with high RNF149 expression. (**B**) Heatmap of the immune infiltration scores of different immunocytes in HCC patients with high or low RNF149 expression based on ssGSEA. (**C**) Comparisons of immune enrichment score and immune cell abundance between RNF149-high and -low expression samples. (**D**) Correlation of RNF149 protein level with M2 macrophage, MDSC, regulatory T cell, and exhausted T cell abundance scores, respectively. ns: *p* ≥ 0.05, ** *p* < 0.01, *** *p* < 0.001.

**Figure 9 cancers-15-05203-f009:**
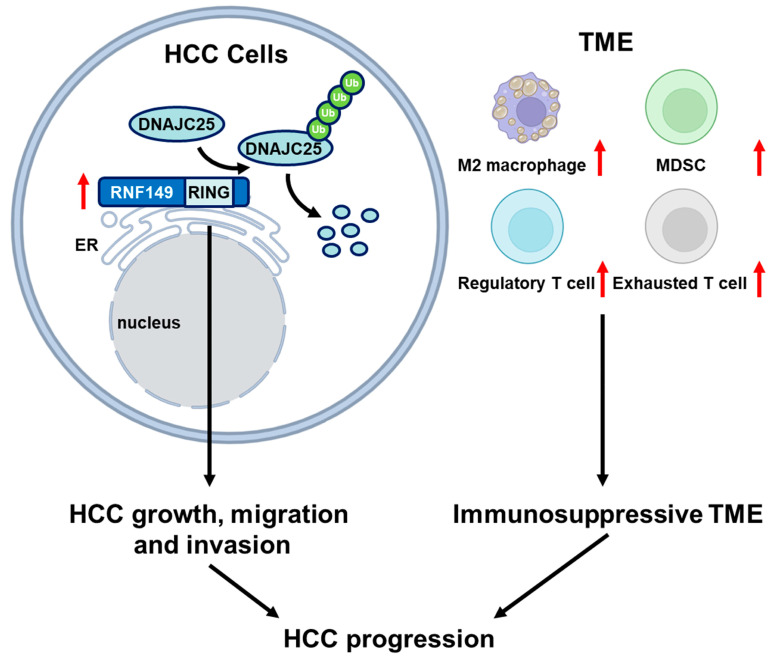
Simplified schematic diagram indicating the potential role of RNF149 in the progression of HCC.

## Data Availability

The data presented in this study are available on reasonable request from the corresponding author.

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
