# Peer review of "RNF149 Promotes HCC Progression through Its E3 Ubiquitin Ligase Activity"

_cancers, 2023, doi:10.3390/cancers15215203_

Round 1

Reviewer 1 Report

This study demonstrated that RNF149 expression correlated
with HCC malignant grades, and RNF149 triggered HCC progression through its ubiquitin ligase activity and thus degrading DNAJC25, but also indicated that RNF149 was associated with TME regulation in HCC.

-The authors should implement the introduction and explain the studies used for these correlations and regulations.

-Details in the materials and methods section should be reported

Minor editing of English language required

Author Response

Response to Reviewers 1:

We thank reviewers for their careful evaluations on our manuscript. In the revised version, we have carefully considered the reviewers’ criticisms and further improved our manuscript.

  1. The authors should implement the introduction and explain the studies used for these correlations and regulations.

Response: Thanks for your suggestions! The clinical relevance of RNF149 and its regulated substrates have been less reported in the previous literature. The relevant literatures were listed in the Introduction section. The data used to analyze the correlation between RNF149 and clinical characteristics of patients and potential regulatory substrates were derived from our team's previous study (Nature, 2019) and was described in detail in the Introduction section. In addition, in the revised manuscript, we added the expression and significance of RNF149 in pan-cancer (Figure S1), which further suggested that the tumor-promoting mechanism of RNF149 through its ubiquitin ligase activity has a relatively wide universality.

Supplementary figure S1.

The expression of RNF149 in pan-cancer.

TCGA transcriptomics data were used to analyze the expression pattern of RNF149. *P < 0.05, ***P < 0.001,****P < 0.001.

  1. Details in the materials and methods section should be reported

Response: According to your suggestion, more details are described in the Materials and methods section which all were highlighted in yellow.

Reviewer 2 Report

With pleasure, I read the paper titled “RNF149 promotes HCC progression through its E3 ubiquitin ligase activity”. The topic is clinically relevant and of importance to the readers of the Cancers journal. Overall, the manuscript reads good and has good flow of ideas, up-to-date citations, and good summary of data using figures. A major strength of the article is being among the first-ever studies to examine the role of RNF149 in HCC. Additional key strengths include the utilization of various cell lines and molecular biology techniques to validate the results. The manuscript is just missing some translational (drug-related) perspectives. Otherwise, the story is pretty solid. I have the following comments:

Introduction. Is RNF 149 upregulated in other cancer types beyond HCC using pan-cancer analysis? Is your study the first ever to examine the oncogenic role of RNF149 in HCC? If so, please highlight this significance in your research.

Figures 1-3. I wonder if the protein and mRNA levels are matched, and therefore, have you investigated the mRNA expression of RNF149 using the same tissues you derived your proteomic data from? Does high protein expression of RNF149 correlate with other prognostic markers, such as degree of differentiation, metastasis, TP53 mutation, MYC amplification, or others?

Figure 4. Please provide colony formation assay results for Huh7 EV vs. RNF149 OE to complement your data in Panel A. examining the supplementary data, the knockdown efficiency for RNF149 in MHCC-97H is not convincing by looking at the western blot itself, despite the quantification showed significant knockdown efficiency. For Panel G, have you examined the protein profile of these xenografts after harvesting for RNF149 and DNAJC25 expression levels?

Figure 5. How was the catalytically dead mutant activity of RNF149 H289A validated? in Panel C, the magnitude of increased proliferation does not seem substantial upon comparing EV or H289A vs. WT.

Figure 6. This is a great figure. Regarding panel H, apart from HEK293T cells, were you able to detect the physical interaction between DNAJC25 and RNF149 endogenously using HCC cell lines?

Figure 7. This figure is great, however, it will be better if you could examine the xenografts in Figure 4, Panel G, for these immune infiltrates to provide actual data from your own study.

General questions that will substantially enhance the quality of your research. If they can be addressed experimentally, this will be great. Otherwise, please provide your thoughts. Have you examined the phenotypic effects of genetic inhibition of RNF149 on DNA damage, cell cycle, EMT, and differentiation. Is there a pharmacologic inhibitor of RNF149 to mimic siRNA knockdown of RNF149? Have you examined if pharmacological inhibition of RNF149 show similar phenotype matched to that of RNF149 knockdown? Does RNF149 overexpression correlated with reduced sensitivity to standard-of-0care chemotherapeutics? The study will become more significant if in-vivo xenograft data are included with respect to pharmacological treatment.

Minor English editing

Author Response

Response to Reviewers:

We thank reviewers for their careful evaluations on our manuscript. In the revised version, we have carefully considered the reviewers’ criticisms and further improved our manuscript.

Reviewer #2: With pleasure, I read the paper titled “RNF149 promotes HCC progression through its E3 ubiquitin ligase activity”. The topic is clinically relevant and of importance to the readers of the Cancers journal. Overall, the manuscript reads good and has good flow of ideas, up-to-date citations, and good summary of data using figures. A major strength of the article is being among the first-ever studies to examine the role of RNF149 in HCC. Additional key strengths include the utilization of various cell lines and molecular biology techniques to validate the results. The manuscript is just missing some translational (drug-related) perspectives. Otherwise, the story is pretty solid. I have the following comments:

Response: We thank the reviewer for his/her positive evaluations on our manuscript. And we have revised our manuscript in the light of the constructive comments.

  1. Introduction. Is RNF149 upregulated in other cancer types beyond HCC using pan-cancer analysis? Is your study the first ever to examine the oncogenic role of RNF149 in HCC? If so, please highlight this significance in your research.

Response: In the light of the constructive suggestion of the referee, we have analyzed the expression pattern of RNF149 using TCGA transcriptomics data. As shown in the Figure S1, in addition to liver hepatocellular carcinoma (LIHC), RNF149 is also significantly upregulated in kidney renal clear cell carcinoma (KIRC), kidney renal papillary cell carcinoma (KIRP), bladder urothelial carcinoma (BLCA), breast invasive carcinoma (BRCA), cholangiocarcinoma (CHOL), esophageal carcinoma (ESCA), head and neck squamous cell carcinoma (HNSC), prostate adenocarcinoma (PRAD), rectum adenocarcinoma (READ), stomach adenocarcinoma (STAD), thyroid carcinoma (THCA) and uterine corpus endometrial carcinoma (UCEC). In addition, our study is the first ever to examine the oncogenic role of RNF149 in HCC based on our review of literature, and the significance has been implemented in the Introduction section of the revised manuscript.

  1. Figures 1-3. I wonder if the protein and mRNA levels are matched, and therefore, have you investigated the mRNA expression of RNF149 using the same tissues you derived your proteomic data from? Does high protein expression of RNF149 correlate with other prognostic markers, such as degree of differentiation, metastasis, TP53 mutation, MYC amplification, or others?

Response: The HCC cohort we used produced matched transcriptomics data, and Pearson correlation analysis showed that RNF149 mRNA expression level was highly consistent with its protein level (correlation coefficient = 0.4, p = 2.5e-07) (Figure S2A).

We further analyzed the mRNA expression of RNF149 in the same HCC cohort. Consistent with the protein level results, RNF149 mRNA was significantly upregulated in tumor tissues and highly expressed in S-III subtypes of HCC (Figure S2B, C). Kaplan-Meier survival analysis also showed that high-expression of RNF149 mRNA was associated with shorter OS and DFS (Figure S2F, G).

The mRNA expression of RNF149 also showed significant upregulation in HCC patients with high serum AFP levels and positively correlated with AFP levels (Figure S2D, E).

Because the degree of differentiation was not provided in the cohort, we only analyzed the correlation between RNF149 protein and mRNA expression and MVI, metastasis, TP53 mutation and MYC amplification. As shown in Figure S3, RNF149 was not significantly associated with those prognostic markers.

  1. Figure 4. Please provide colony formation assay results for Huh7 EV vs. RNF149 OE to complement your data in Panel A.

Response: The colony formation assay results for Huh7 EV vs. RNF149 OE was shown in Figure 4C, which validated that overexpression of RNF149 significantly promoted colony formation of HCC cells.

  1. Figure 4. Examining the supplementary data, the knockdown efficiency for RNF149 in MHCC-97H is not convincing by looking at the western blot itself, despite the quantification showed significant knockdown efficiency.

Response: The knockdown efficiency for RNF149 in MHCC-97H has been reconfirmed by WB in new Figure S4B.

  1. Figure 4. For Panel G, have you examined the protein profile of these xenografts after harvesting for RNF149 and DNAJC25 expression levels?

Response: According to your good suggestion, we detected the protein expression of RNF149 in these xenografts after harvesting. As shown in new Figure 5B, RNF149 expression was upregulated in RNF149-overexpressed tumors compared with control samples. As for DNAJC25, we have try some antibodies for DNAJC25 from Bioss ( bs-14389R ) and Thermo Fisher Scientific ( PA5-69340), and unfortunately, we could not successfully detect DNAJC25 protein. Therefore, we are sorry for not providing the expression level data of DNAJC25 in subcutaneous tumors.

  1. Figure 5. How was the catalytically dead mutant activity of RNF149 H289A validated?

Response: The histidine residue (e.g., RNF149 H289) in the RING domain is well conserved among RING domains of E3 ligases and is essential for accepting ubiquitin from E2[1]. More importantly, according to the Hong SW et al’ s study[2], RNF149 H289A mutant displayed as an inactive E3 ubiquitin ligase in cells, which was validated by the ubiquitination level and protein level of its substrate BRAF. Thus, we used the RNF149 H289A mutant as the catalytically dead mutant. The two references were also cited in our revised manuscript.

  1. Figure 5. In Panel C, the magnitude of increased proliferation does not seem substantial upon comparing EV or H289AVs.WT.

Response: As shown in Figure 6C, overexpression of RNF149 promoted the colony formation of HCC cells compared with that of EV-control and RNF149 H289A-overexpressing cells. The fold change of the colony-formation-promoting effect is approximately 1.5, and further confirmed by statistics test with P value<0.01. Notably, similar with EV control, RNF149 H289A mutant did not promote colony formation of HCC cells, which indicated that RNF149 promoted colony formation of HCC cells through its ligase activity.

  1. Figure 6. This is a great figure. Regarding panel H, apart from HEK293T cells, were you able to detect the physical interaction between DNAJC25 and RNF149 endogenously using HCC cell lines?

Response: As mentioned in the answer for your Question 5, although this experiment is exactly what we have always wanted to do, we cannot detect the endogenous interaction between RNF149 and DNAJC25, because of the lack of the effective antibodies for DNAJC25.

  1. Figure 7. This figure is great, however, it will be better if you could examine the xenografts in Figure 4, Panel G, for these immune infiltrates to provide actual data from your own study.

Response: We thank your good suggestion. Due to the NOD-SCID mouse used in xenograft experiments were severely immunodeficient, we might not be able to detect the potential immune infiltrates in the xenografts. Further, we will take well-designed animal experiments for the determination of RNF149 functions in immune regulation of HCC.

  1. Have you examined the phenotypic effects of genetic inhibition of RNF149 on DNA damage, cell cycle, EMT, and differentiation.

Response: According to your suggestion, we examined the changes of EMT markers after knocking down RNF149 in MHCC-97H and PLC cells, respectively. Our results showed that after knocking down RNF149, N-Cadherin was significantly down-regulated in MHCC-97H cells and Vimentin was significantly down-regulated in PLC cells- (Figure S5), indicating that RNF149 might promote EMT, which is consistent with our hypothesis. In addition, our previous analysis based on proteomics profiling data from HCC cell lines indicated that HCC cells with overexpression or knockdown of RNF149 showed significantly upregulated or downregulated proteins enriched in G2/M transition, regulation of stem cell differentiation, while proteins associated with DNA damage showed no significant enrichment (Figure 3D). Due to time limit (10 days), we did not obtain the experiment results of the correlated cellular phenotype on cell cycle, differentiation, and we will further experimentally validate the effects of RNF149 on those phenotypes in the future.

  1. Is there a pharmacologic inhibitor of RNF149 to mimic siRNA knockdown of RNF149? Have you examined if pharmacological inhibition of RNF149 show similar phenotype matched to that of RNF149 knockdown? Does RNF149 overexpression correlated with reduced sensitivity to standard-of-care chemotherapeutics? The study will become more significant if in-vivo xenograft data are included with respect to pharmacological treatment.

Response: We thank your valuable comment. Through searching for specific pharmacologic inhibitor of RNF149 in public databases including DrugBank and associated literatures, we did not find a targeted inhibitor of RNF149, and we sorry for provided the correlated results.

To investigate whether the change of RNF149 protein level affects the sensitivity of standard-of-care chemotherapeutics, the IC50 of sorafenib after overexpression or knockdown of RNF149 in Huh7 or MHCC-97H cells was detected. The results showed that the IC50 of sorafenib increased after overexpression of RNF149 and decreased after knockdown of RNF149, which is consistent with our expectation (Figure 4E).

  1. Deshaies, R.; Joazeiro, C. RING domain E3 ubiquitin ligases. Annual review of biochemistry 2009, 78, 399-434, doi:10.1146/annurev.biochem.78.101807.093809.
  2. Hong, S.-W.; Jin, D.-H.; Shin, J.-S.; Moon, J.-H.; Na, Y.-S.; Jung, K.-A.; Kim, S.-M.; Kim, J.C.; Kim, K.-p.; Hong, Y.S.; et al. Ring Finger Protein 149 Is an E3 Ubiquitin Ligase Active on Wild-type v-Raf Murine Sarcoma Viral Oncogene Homolog B1 (BRAF). Journal of Biological Chemistry 2012, 287, 24017-24025, doi:10.1074/jbc.M111.319822.

Reviewer 3 Report

Hepatocellular carcinoma (HCC) accounts for over 80% of cases among liver cancer with high incidence and poor prognosis. Thus, it is of valuable clinical significance for discovery of potential biomarkers and drug targets for HCC. Herein, based on proteomics data analysis, RNF149 was found to be elevated in hepatocellular carcinoma (HCC) tissues and associated with HCC malignancy, which was validated by immunohistochemistry (IHC) staining. RNF149 was demonstrated to promote proliferation, migration and invasion of HCC cells dependent on its E3 ubiquitin ligase activity in vitro, and further bioinformatics analysis indicated that highexpression of RNF149 correlated with immunosuppressive tumor microenvironment (TME). These results suggest that RNF149 could exert pro-tumor functions in HCC in dependence of its E3 ubiquitin ligase activity and might be a potential prognostic marker and therapeutic target for HCC treatment. But I have several following concerns:

1. Abbreviations should be defined When thery appears in the first time. Such as "RNF149".

2. Lines 88-91, The reference should not appear in the middle of the sentence, and the two verbs of the sentence are not unified (non-parallel construction).

3. Line 94, there should be a blank between "R" and "(".

4. The nucleic acid sequences (including gene names, regulatory sequences, and primer names) should be in italics

5. The "P" representing the statistical difference should be italicized.

6. Please draw a diagram summarizing the signaling pathway of RNF149-mediated HCC progression to help readers understand the content of the authors' research visually.

7. Please unify the format of references in the article, including the author's name, the case of words in the title of the article, the writing of the name of the journal, and the page number.

Minor editing of English language required.

Author Response

Response to Reviewers:

We thank reviewers for their careful evaluations on our manuscript. In the revised version, we have carefully considered the reviewers’ criticisms and further improved our manuscript.

Reviewer #3:

  1. Abbreviations should be defined When they appear in the first time. Such as "RNF149".

Response: Thanks for your suggestion. We have checked the first appearance of all abbreviations with corresponding definition, and improved our manuscript accordingly.

  1. Lines 88-91, The reference should not appear in the middle of the sentence, and the two verbs of the sentence are not unified (non-parallel construction).

Response: We sorry for the mistake, and have corrected the mistakes in the revised manuscript.

  1. Line 94, there should be a blank between "R" and "(".

Response: The blank was added in the revised manuscript.

  1. The nucleic acid sequences (including gene names, regulatory sequences, and primer names) should be in italics

Response: The nucleic acid sequences (including gene names, regulatory sequences, and primer names) in revised manuscript have been corrected in italics.

  1. The "P" representing the statistical difference should be italicized.

Response: The "P" representing the statistical difference in revised manuscript has been corrected in italics.

  1. Please draw a diagram summarizing the signaling pathway of RNF149-mediated HCC progression to help readers understand the content of the authors' research visually.

Response: Thanks for the constructive comment. We provided a schematic diagram as new Figure 9.

  1. Please unify the format of references in the article, including the author's name, the case of words in the title of the article, the writing of the name of the journal, and the page number.

Response: We unified the format of references in the revised manuscript.

Round 2

Reviewer 2 Report

The authors did a great job by attending to almost all comments adequately. The limitations were clearly acknowledged. The manuscript now reads well and is methodologically robust, scientifically sound, and intellectually curious. Well-done. I recommend the acceptance of the manuscript in its current form.

None to minor English editing may be needed.

Reviewer 3 Report

The authors have addressed all my concerns. I recommend accepting this manuscript in current form.